# Real-Time High-Level Acute Pain Detection Using a Smartphone and a Wrist-Worn Electrodermal Activity Sensor

**DOI:** 10.3390/s21123956

**Published:** 2021-06-08

**Authors:** Youngsun Kong, Hugo F. Posada-Quintero, Ki H. Chon

**Affiliations:** Biomedical Engineering Department, University of Connecticut, Storrs, CT 06269, USA; youngsun.kong@uconn.edu (Y.K.); hugo.posada-quintero@uconn.edu (H.F.P.-Q.)

**Keywords:** pain, electrodermal activity, smartphone, machine learning

## Abstract

The subjectiveness of pain can lead to inaccurate prescribing of pain medication, which can exacerbate drug addiction and overdose. Given that pain is often experienced in patients’ homes, there is an urgent need for ambulatory devices that can quantify pain in real-time. We implemented three time- and frequency-domain electrodermal activity (EDA) indices in our smartphone application that collects EDA signals using a wrist-worn device. We then evaluated our computational algorithms using thermal grill data from ten subjects. The thermal grill delivered a level of pain that was calibrated for each subject to be 8 out of 10 on a visual analog scale (VAS). Furthermore, we simulated the real-time processing of the smartphone application using a dataset pre-collected from another group of fifteen subjects who underwent pain stimulation using electrical pulses, which elicited a VAS pain score level 7 out of 10. All EDA features showed significant difference between painless and pain segments, termed for the 5-s segments before and after each pain stimulus. Random forest showed the highest accuracy in detecting pain, 81.5%, with 78.9% sensitivity and 84.2% specificity with leave-one-subject-out cross-validation approach. Our results show the potential of a smartphone application to provide near real-time objective pain detection.

## 1. Introduction

Acute pain is the most common type of pain that anyone can experience. A lot of efforts have been made by many researchers to assess acute pain to provide precise treatments to patients [1,2,3]. However, the essential problem of pain assessment is that pain perception is subjective, making it difficult for patients to correctly describe their symptoms to healthcare providers. This may lead to incorrect prescriptions for higher doses of medications, which can result in drug overuse and addiction [4,5,6]. According to the Centers for Disease Control and Prevention (CDC), there were nearly 47,000 overdose deaths in the U.S. involving opioids in 2018, and almost one third involved prescription opioids [7]. Moreover, the economic burden caused by opioids was alleged to be about $78.5 billion in 2013 [8]. Therefore, there is a need for a way to objectively measure pain, to aid acute pain patients and healthcare providers to define more accurate treatments and prescription doses.

Given that pain events can be observed in patients’ homes, ambulatory measurement based on smartphones can greatly contribute to home treatments. Rather than developing dedicated devices for pain monitoring, a smartphone application can use the phone’s powerful processors to provide real-time data analysis. There have already been a few pain management applications for smartphone users, however, none of them offer both real-time and objective pain measurement [9,10]. Hasan et al. developed a pain detection smartphone application using facial expression captured by the phone [11]. Although they provided an objective measure of pain, their application showed optimal performance only for machine-learning models exclusively trained for each individual (i.e., longitudinal study) and has some limitations for practical use. First, facial expressions are easy to manipulate (i.e., the application can be abused), and the application cannot be used for ambulatory recording as it is difficult to take continuous photos of the face. All other smartphone applications are based on self-reported criteria (e.g., 0–10 pain scale). Thus, the need for a smartphone application that can objectively measure pain in real-time is currently unmet in practice.

To measure pain in real-time, a physiological signal that can be measured continuously is necessary. As pain elicits a sympathetic nervous system (SNS) reaction, several physiological signals can be considered, including electrodermal activity (EDA), photoplethysmography (PPG), and electrocardiography (ECG). EDA can capture the sympathetic reactivity to pain and has shown to be sensitive to pain [12,13]. PPG and ECG can be used to assess autonomic arousal via heart rate variability (HRV) analysis, but it is not as accurate as EDA for providing assessment of the sympathetic nervous system [14]. EDA measures the sweat gland activities affected by the sympathetic innervation, so it is often measured from palms and fingers where the density of sweat glands is higher. Although EDA has shown sensitivity to pain [14], none of the EDA indices were developed for real-time applications. To this end, we have developed an objective real-time pain measuring algorithm and implemented it in a smartphone application using an EDA wearable device. We tested the hypothesis that pain can be detected in real-time by an EDA device connected to a smartphone application. Unlike other smartphone applications for pain management, our implementation measures pain based on objective methods in near real-time, which is more conducive to ambulatory monitoring. The accuracy of objective pain detection using a smartphone application was enhanced by the use of machine learning with features derived from EDA. A preliminary version of this work has been reported [15].

## 2. Methods

### 2.1. EDA Features

Previous studies have found that time-frequency spectral analysis can provide a good degree of sensitivity to pain [16,17,18,19]. We recently found that time-frequency spectral analysis can provide more sensitive to the sympathetic activity including pain when compared to the traditional features (e.g., phasic and tonic component of EDA) [14,15,20,21]. Moreover, we found that the differential characteristic features derived from traditional features (e.g., phasic component) and the time-frequency spectral analysis features also showed good performance to detect pain [22]. However, we only selected the most discriminative EDA features into our smartphone application since it has limited computing processing speed for real-time processing. Among the several features, we only chose derivative of phasic component of EDA (dPhEDA) derived from phasic components, and spectral features time-varying index of sympathetic activity (TVSymp) and modified time-varying index of sympathetic activity (MTVSymp) [14,15,22], as they use VFCDM, a technique shown to exhibit one of the highest time- and frequency-spectrum resolutions with accurate amplitude when compared to other similar methods (e.g., Wigner–Ville and continuous wavelet transform) [23].

First, the EDA signals were preprocessed as follows: (1) we resampled to 4 Hz from 130 Hz, (2) we applied a median filter with a 1-sec window, (3) we resampled to 2 Hz, and (4) we applied a highpass filter with a cutoff frequency at 0.01 Hz. From the preprocessed EDA, TVSymp and dPhEDA were calculated, as depicted in Figure 1. The reason we calculated these three different features was to examine which of the three provides the best quantitative assessment of pain. The mathematical details of these features are described below.

#### 2.1.1. Time-Varying Index of Sympathetic Activity (TVSymp) and Modified TVSymp (MTVSymp)

TVSymp has shown its sensitivity to pain [14]. The TVSymp computation consists of two parts: (1) variable frequency complex demodulation (VFCDM) to reconstruct EDA signals with components in the range of 0.08–0.24 Hz [23], and (2) Hilbert transform to obtain instantaneous amplitude of the reconstructed signals. Since TVSymp is thoroughly described in a previous publication [14], we briefly summarize TVSymp computation in this section. First, the EDA signal *x*(*t*) can be considered to be a narrow band oscillation with a center frequency *f*_0_, instantaneous amplitude *A*(*t*), phase *φ*(*t*), and the direct current component *dc*(*t*), as follows:(1)$$x\left(t\right)=dc\left(t\right)+A\left(t\right)\cos\left(2\pi f_{0}t+\varphi \left(t\right)\right).$$

For a given center frequency, the instantaneous amplitude information *A*(*t*) and phase information *φ*(*t*) can be extracted by multiplying Equation (1) by $e^{-j2\pi f_{0}t}$, which results in the following:(2)$$z\left(t\right)=dc\left(t\right)e^{-j2\pi f_{0}t}+\frac{A\left(t\right)}{2}e^{j\varphi \left(t\right)}+\frac{A\left(t\right)}{2}e^{-j\left(4\pi f_{0}t+\varphi \left(t\right)\right)}.$$

By shifting $e^{-j2\pi f_{0}t}$ to the left, the center frequency *f*_0_ moves to zero frequency in the spectrum of *z*(*t*). If *z*(*t*) is subjected to an ideal low-pass filter (LPF) with a cutoff frequency *f*_c_ < *f*_0_, then the filtered signal $z_{lp}\left(t\right)$ will contain only the component of interest, and we can obtain the following equations:(3)$$z_{lp}\left(t\right)=\frac{A\left(t\right)}{2}e^{j\varphi \left(t\right)},$$
(4)$$A\left(t\right)=2\left|z_{lp}\left(t\right)\right|,$$
(5)$$\varphi \left(t\right)=arctan\left(\frac{\mathrm{imag}\left(z_{lp}\left(t\right)\right)}{\mathrm{real}\left(z_{lp}\left(t\right)\right)}\right).$$

In the case that the modulating frequency is not fixed but varies as a function of time, the signal *x*(*t*) can be expressed as follows:(6)$$x\left(t\right)=dc\left(t\right)+A\left(t\right)\left(\int_{0}^{t}\cos\left(2\pi f\left(\tau \right)d\tau +\varphi \left(t\right)\right)\right).$$

Similar to Equations (1) and (2), multiplying Equation (6) by $e^{-j\int_{0}^{t}2\pi f\left(\tau \right)d\tau }$ produces both instantaneous amplitude, *A*(*t*), and instantaneous phase *φ*(*t*), as follows:(7)$$z\left(t\right)=x\left(t\right)e^{-j\int_{0}^{t}2\pi f\left(\tau \right)d\tau }=dc\left(t\right)e^{-j\int_{0}^{t}2\pi f\left(\tau \right)d\tau }+\frac{A\left(t\right)}{2}e^{j\varphi \left(t\right)}+\frac{A\left(t\right)}{2}e^{-j\int_{0}^{t}4\pi f\left(\tau \right)d\tau }.$$

From Equation (7), by applying an ideal LPF to *z*(*t*) with a cutoff frequency *f*_c_ < *f*_0_, the filtered signal $z_{lp}\left(t\right)$ can be obtained with the same instantaneous amplitude *A*(*t*) and phase *φ*(*t*) as provided in Equations (4) and (5). The instantaneous frequency is given by:(8)$$f\left(t\right)=f_{0}+\frac{1}{2\pi }\frac{d\varphi \left(t\right)}{dt}$$

TVSymp uses 2 Hz EDA signals, and VFCDM with 2 Hz of sampling frequency decomposed the signals with centered spectral frequencies from 0.04 to 0.92 Hz by stepping through at 0.08 Hz increments. We summed the second and third components to include the sympathetic dynamics, which range between 0.045–0.25 Hz, followed by normalization to unit variance. The summed value is denoted by *X*’. Its instantaneous amplitude is then computed using the Hilbert transform as follows:(9)$$Y^{\prime }\left(t\right)=\frac{1}{\pi }p.v\int_{-\infty }^{\infty }\frac{X\prime \left(\tau \right)}{t-\tau }d\tau$$
where *p*.*v* represent the Cauchy principal value. As *X*’(*t*) and *Y*’(*t*) form the complex conjugate pair, an analytic signal, *Z*(*t*), can be defined as follows:(10)Z(t)=X′(t)+iY′(t)=a(t)ejθ(t)a(t)=[X′2(t)+Y′2(t)]12θ(t)=arctan(Y′(t)/X′(t))

Finally, TVSymp, *a*(*t*), is obtained by calculating the instantaneous amplitude of *Z*(*t*). We then calculated MTVSymp based on TVSymp to emphasize EDA changes caused by pain and remove other baseline EDA responses from the prior segments. Each time point of MTVSymp is calculated by subtracting the average of samples corresponding to *k*-seconds back each time point of TVSymp, and setting it to zero if the averaged value is greater than TVSymp. MTVSymp’s equation can be shown as follows:(11)$$MTVSymp_{t}=\{\begin{array}{r} a_{t}-\mu_{t},\;\;\mu_{t}\leq a_{t}\\ 0,\;\;\mu_{t}>a_{t} \end{array},\;\mu_{t}=\frac{1}{k\cdot F_{s}}\sum_{i=t-k\cdot F_{rs}}^{t-1}a_{i},$$
where *k* and $F_{s}$ represent a length of time window and the sampling frequency (2 Hz in the paper), respectively. The length of time window was set to 5 s in order to reflect the most immediate changes of EDA due to pain stimulus and to minimize the loss of information since pain induced EDA dynamics happen rather quickly. In addition, this choice of time window was based on the purpose to provide near real-time analysis of pain. Greater time window may yield better results but we also wanted good pain detection in near real-time. 

#### 2.1.2. Derivative of Phasic Component of EDA (dPhEDA)

We describe the derivative of the phasic component of EDA (dPhEDA). EDA consists of tonic and phasic components, which represent slow and fast dynamics, respectively. First, we used the convex optimization approach (cvxEDA) to decompose EDA signals into phasic and tonic components [24]. We then applied the five-point stencil central finite differences equation [25] as follows:(12)$$dphEDA_{n}=\frac{{x_{phasic}}_{n-2}-8\cdot {x_{phasic}}_{n-1}+8\cdot {x_{phasic}}_{n+1}-{x_{phasic}}_{n+2}}{12\cdot \left(1/F_{s}\right)},$$
where $x_{phasic}$ and $F_{s}$ represent a processed phasic component extracted from EDA signals and the sampling frequency (2 Hz), respectively.

### 2.2. Smartphone Application Development

Figure 2 shows the graphical user interface of our smartphone application and its unified model language (UML) sequence diagram. The application has dedicated buttons to connect to a wearable device, start and stop recording, and end the application. The application graphs the raw EDA in the top graph and the analysis based on TVSymp, MTVSymp, and dPhEDA in the bottom graph in near real-time. Our application remotely collects EDA signals via the Bluetooth protocol from an EDA wearable device (Shimmer 3, Shimmer, Dublin, Ireland) and calculates EDA features in near real-time [26]. The EDA features are TVSymp, MTVSymp, and dPhEDA, as described in the previous section. We used Java and C++ with Java native interface (JNI) for the user interface and our signal processing techniques, respectively. The Shimmer 3 Java/Android application programming interface (API) was used for connection between our application and the wearable device, Shimmer 3. The Eigen 3 library was used for matrix arithmetic operations and numerical solvers [27]. The application collects unfiltered EDA signals, calculates EDA features, shows both 30-s raw and calculated signals, and saves them in a text file with timestamps (Figure 2a,b). Although EDA signals are transmitted to the smartphone at ~120 Hz, EDA indices are generated at around 15 Hz due to wireless communication and computational loads at the smartphone processor. Due to the varying sampling frequency, we also used the cubic spline algorithm to resample at 4 Hz.

The application calculates three different time series from which we obtain pain features. Figure 3 shows our scheme to apply to a time series for real-time monitoring. We applied a different number of samples for TVSymp and dPhEDA computations due to performance limitations of the smartphone’s processor. By using the trial and error approach, we confirmed that 55- and 25-s windows allowed us a good balance between accuracy and computational load for obtaining real-time TVSymp and dPhEDA, respectively, using a smartphone. To compensate for the first 55 and 25 s of EDA recording, we padded the first five seconds with the average value of the time series. Additionally, we padded the last value of EDA signals 20 times (5 s times 4 Hz) at the end of each signal to avoid the corruption of TVSymp and dPhEDA time series. After calculating the time series, we averaged the last 2 s to compute the features (except for the 5-s padded signals) and then appended this average to the end of our real-time features. An example of some obtained time series is shown in Figure 4.

### 2.3. Experiments

We conducted two experiments. The first was to collect data using a wrist-worn EDA device with thermal pain, and the second was using a lab device (i.e., non-wearable device) with electrical pain. There were no duplicate subjects between electrical and thermal pain experiments. All experiments were carried out in a quiet room to minimize any other stressors. For both experiments, we used our smartphone application to process EDA signals. The EDA signals were collected and processed at the same time for the former experiment. For the latter experiment, the pre-collected data were continuously sent to a smartphone application in real-time using a data threading technique and then another thread for streaming data was used to calculate EDA features in real-time. Our protocols were approved by the University of Connecticut Institutional Review Board (IRB).

#### 2.3.1. Thermal Grill Experiment

First, we recruited ten subjects (4 females and 6 males, 28.9 ± 4.7 years old) to collect EDA signals with high levels of heat pain using a thermal grill (TG). TG, demonstrated in 1896 by Thunberg, has been widely exploited in pain research [28,29,30,31]. For example, researchers have found that TG pain, which is induced by combining both cold and hot water, leads to sensation of heat pain largely due to inhibition of the cold pain receptors [29]. In addition, other studies have suggested that TG pain perception may be related to neuropathic pain [32,33]. Therefore, a TG is an effective tool to induce heat pain without any damaging tissue injury, and can safely induce even high levels of heat pain [34]. A TG is made of alternating copper pipes which run cold water in one direction and warm water through every-other pipe in the same direction, as shown in Figure 5. The cold water was maintained to ~18 °C with ice, and the warm water was set to between 50–58 °C using a feedback-controlled warm water bath (Isotemp GPD 2S, Fisher Scientific, Waltham, MA, USA). The sensed temperature observed through a thermal camera was ~35 °C (FLIR One, Wilsonville, OR, USA). For each subject, the warm water’s temperature was set based on it inducing a level 8 out of 10 on the visual analogue scale (VAS) pain score; hence, we designate this as high-level pain. The greater the temperature difference between the warm and cold water, the higher the level of pain perceived by the subjects. We then asked each subject to put their right hands on the TG until the pain perception became unbearable. This procedure was repeated 10 times per subject, with random intervals. Shimmer 3 (Shimmer, Dublin, Ireland) was used to collect EDA. The electrodes were placed on the index and middle fingers of the subjects’ left hands. The device transmitted the EDA signals via Bluetooth to a Galaxy S10 smartphone (Samsung, Seoul, Korea) placed within 0.5 m from the EDA device. The smartphone received and processed the EDA signals using its own processors during data collection.

#### 2.3.2. Electrical Pulse Experiment

We also recruited 15 subjects (9 females and 6 males, 25.6 ± 4.8 years old) to collect EDA signals with high levels of pain induced by electrical pulse (EP). EDA was collected from index and middle fingers using a galvanic skin response device (GSR MP 160) and amplifier (BIONAMADIX 2CH Amp). The EDA signals were collected at 1000 Hz using the AcqKnowledge software (BIOPAC Systems, Inc. Goleta, CA, USA). EP stimuli were given using STMISOC (BIOPAC Systems, Inc. Goleta, CA, USA). For each subject, we first found the personalized stimulus level that induced a level 7 out of 10 VAS pain score by adjusting the electrical pulse amplitude (pulse width was fixed to 10 ms). This high level of EP was inflicted 10 times with random intervals. Note that we used multithreading techniques on the smartphone application to simulate real-time data processing. The collected data were sampled at 25 Hz on the smartphone, which processed the data using its own processors.

### 2.4. Statistics

In order to compare pain and painless segments, we first extracted painless and pain segments from EDA features obtained using the smartphone application. We considered 5 s before and after each pain stimulus to be painless and pain segments, respectively. To test the feasibility of pain detection with our real-time EDA indices, we evaluated features based on linear discriminating power, overall classification power, and significance of differences between painless and pain segments, using Fisher’s ratio, area under the receiver operating characteristics (AUROC), and statistical hypothesis tests, respectively. Fisher’s ratio estimates the linear discriminating power between two variables using the mean and variance [35]. We used the following equation:(13)$$FiR_{i}=\frac{\left|{\bar{X_{i}}}^{\left(0\right)}-{\bar{X}}_{i}^{\left(1\right)}\right|}{\sqrt{var{\left(X_{i}\right)}^{\left(0\right)}+var{\left(X_{i}\right)}^{\left(1\right)}}},$$
where *X_i_*(*n*) represents the *i*^th^ feature’s class n. The higher the Fisher’s ratio, the stronger the discriminating power between two classes. AUROC also estimates the degree of discrimination between two classes [36]. By calculating AUROC, we can observe the classification power of each feature in the classifiers. ROC curves are obtained by calculating true positive and false positive rates by adjusting classification thresholds, which ranges between 0–1. Finally, we fitted linear mixed effects models (R-package) for normally distributed values and used the nested Ranks Test (R-package) for non-normally distributed values to reflect variance changes within each subject [37,38,39]. Normality was tested using the Kolmogorov–Smirnov test [40].

### 2.5. Machine Learning

We performed machine learning using Python 3.6 and Scikit learn packages [41] to examine if our calculated features can detect induced pain. We used three different protocols when using both datasets together: (1) leave-one subject out cross-validation for both EP and TG datasets, (2) training with EP and testing with TG, and (3) training with TG and testing with EP. Note that there were no duplicate subjects between TG and EP experiments. Eight different classifiers were tested, which consisted of support vector machine (SVM) with linear activation function (L-SVM), 3rd order polynomials (P-SVM), and radial basis function kernels (R-SVM); a decision tree (DT); random forest (RF); multi-layer perceptron (MLP); logistic regression (LR); and K-nearest neighbors (KNN). Except for the tree-based classifiers, all classifiers were used after data standardization with zero means and unit variance. Note that we did not apply any data balancing techniques as the dataset was already balanced.

We optimized parameters of each classifier using the grid search cross-validation technique with a 5-fold (i.e., subject-wise 5-fold) cross validation, as shown in Figure 6. We excluded test datasets for each fold. Scoring metrics for parameter optimization were cross-entropy loss for MLP and the accuracy for others. C and gamma were optimized for SVM, and the criterion function was optimized for decision tree and random forest. For MLP, the number of hidden layers was chosen between 1–3 with 100 hidden units per layer. Additionally, the number of epochs was fixed to 100. The activation function, solver, and the initial learning rate were optimized. Further, the learning rate for the stochastic gradient descent (SGD) solver was adjusted by the division of 5 each time two consecutive epochs failed to show improvement on the validation set. For logistic regression and KNN, solver and K were chosen based on the grid search technique. The details of optimized parameters are described in Table 1. We calculated accuracy, sensitivity, and specificity to evaluate the classifiers as follows:(14)$$\mathrm{Accuracy}=\frac{TP+TN}{TN+FP+FN+TP},$$
(15)$$\mathrm{Sensitivity}=\frac{TP}{TP+FN},\;$$
(16)$$\mathrm{Specificity}=\frac{TN}{TN+FP},\;$$
where *TP*, *TN*, *FP*, and *FN* represent true positive, true negative, false positive, and false negative, respectively. For leave-one-subject-out cross-validation, we averaged accuracy, sensitivity, and specificity across all folds.

## 3. Results

Figure 7 shows a comparison of VAS between EP and TG. Mean ± standard deviation of EP and TG were 6.20 ± 1.49 and 7.86 ± 0.70. TG showed higher mean of VAS and lower standard deviation of VAS than those of EP. Significant difference was observed between VAS of EP and TG (*p* = 0.0019, linear mixed effects model; R-package) [38,39].

Table 2 shows the statistical analysis results on the difference between painless and pain segments for both datasets. The maximum value of each feature showed higher Fisher’s ratio and AUROC when compared to their mean value, except for dPhEDA. MTVSymp showed higher Fisher’s Ratio and AUROC than did the two other features. The mean value of dPhEDA showed higher Fisher’s ratio than that of TVSymp, while the maximum value of TVSymp showed higher Fisher’s ratio than did dPhEDA. Both mean and the maximum value of dPhEDA showed higher AUROC than that of TVSymp. All features showed significant difference between painless and pain segments (*p* < 0.001). Figure 8 shows boxplots of the maximum values of TVSymp, MTVSymp, and dPhEDA from both EP and TG datasets.

Table 3 shows the accuracy, sensitivity, and specificity of classifiers trained and tested with three different protocols for both TG and EP datasets. When using both datasets with leave-one-subject-out cross-validation, random forest showed the highest accuracy of 81.5% with 78.9% and 84.2% sensitivity and specificity, respectively, followed by logistic regression with 81.3%, 75.4%, and 87.3% accuracy, sensitivity, and specificity, respectively. Mean ± standard deviation of logistic regression classifiers’ coefficients were found to be: −0.40 ± 0.08, 0.31 ± 0.16, −0.49 ± 0.12, 2.93 ± 0.17, 1.26 ± 0.11, and −0.52 ± 0.09 for TVSymp mean, TVSymp max., MTVSymp mean, MTVSymp max., dPhEDA mean, dPhEDA max., respectively. Moreover, L-SVM and R-SVM showed >80% accuracies. When testing TG using classifiers trained with EP (i.e., protocol 2), MLP and logistic regression showed 80% accuracies. When testing EP with TG-trained classifiers (i.e., protocol 3), L-SVM, R-SVM, random forest, logistic regression, and KNN showed greater than 80% accuracies. 

### 3.1. Electrical Pulse

Table 4 shows the statistical analysis results on the difference between painless and pain segments for the EP dataset. All maximum values of each feature showed higher Fisher’s ratio and AUROC than did their mean values. MTVSymp and dPhEDA exhibited higher Fisher’s Ratio and AUROC when compared to TVSymp. All features showed significant difference between painless and pain segments (*p* < 0.001), except for the mean of TVSymp (*p* = 0.0731). Figure 9 shows boxplots of the maximum values of TVSymp, MTVSymp, and dPhEDA from the EP dataset.

Except for decision tree, all classifiers showed more than 80% of accuracy, as shown in Table 5. Random forest showed the highest accuracy of 84.3% with 87.7% and 80.9% sensitivity and specificity, respectively, followed by KNN with 84.1% accuracy.

### 3.2. Thermal Grill

Table 6 shows the statistical analysis results on the difference between painless and pain segments for the TG dataset. All maximum values of each feature showed higher Fisher’s ratio and AUROC than did their mean values. MTVSymp showed higher Fisher’s Ratio than did other features, followed by TVSymp. For AUROC, dPhEDA exhibited the best performance when compared to the two other features, while MTVSymp showed the poorest performance. TVSymp and MTVSymp showed significant difference between painless and pain segments (*p* < 0.001). Figure 10 shows boxplots of the maximum values of TVSymp, MTVSymp, and dPhEDA from the TG dataset.

Table 7 shows machine-learning results for the TG dataset. Support vector machine with 3rd order polynomial kernel showed the highest accuracy of 76.5% with 61.0% and 92.0% sensitivity and specificity, respectively, followed by logistic regression with 76.0% accuracy. All classifiers exhibited higher specificity than sensitivity.

## 4. Discussion

We developed an objective real-time pain measuring scheme using three EDA indices—TVSymp, MTVSymp, and dPhEDA—and embedded them in a smartphone application using the smartphone’s processors. We induced thermal and electrical pain and collected EDA data using wearable and non-wearable devices, respectively. The EDA indices analyzed using a smartphone exhibited significant differences between induced pain and no pain. In our previous study, we showed that the differential characteristics of EDA signals (dPhEDA and MTVSymp) showed high sensitivity to detecting electrical pain [22]. In this work, MTVSymp and dPhEDA exhibited good sensitivity to both thermal and electrical pain with higher Fisher’s ratio and AUROC than that of TVSymp. Hence, both MTVSymp and dPheEDA features were founded to be important in detecting acute pain. Overall, MTVSymp was the best approach in discriminating between pain and no pain. Furthermore, our machine-learning results showed that induced pain can be detected with up to 81.5% accuracy when both datasets (TG and EP) were used. Moreover, training with a different pain dataset (e.g., testing of EP based on TG training and vice versa), we found 80.0% and 83.1% accuracies for TG and EP, respectively. These results show the potential for other types of pain to be detected using our smartphone application. 

With the promising performance of pain detection using smartphone-computed EDA indices, we can maximize the advantages of a smartphone as it is especially attractive for home and small clinic use. First, we can readily build machine-learning classifiers into our smartphone application without degrading its performance, as calculations to train the classifiers involve simple matrix arithmetic (mostly multiplication and addition), which is computationally inexpensive. For example, dedicated smartphone applications can be developed for specific disorders that cause different types of pain, such as back pain, with further training data. In addition, we can leverage one of the best advantages of smartphones, which is their communication ability. For example, patients can share their pain data immediately with their healthcare providers, which will be helpful for those who need urgent care. Based on our method of providing the core technology, various kinds of smartphone applications can be developed for different purposes and other pain related diseases. 

There are many pain management applications available based on patients’ subjective pain levels [10,47,48,49]. They provide personalized pain treatment and assessment strategies by asking several questions, including pain location, pain symptoms, pain levels, or demographic information. Among these, only our smartphone application measures pain using an objective measurement (i.e., EDA), while others use subjective pain level scales such as numeric rating scales or the visual analog scale. More effective pain treatment and assessment could be developed based on the combination of our objective tool with other pain management applications. Our smartphone’s ambulatory recording can be useful to improve the currently available mobile applications for pain assessment.

Our smartphone application showed some promise of objective pain detection in near real-time. It also has the potential to provide continuous assessment of pain in both home and clinical settings. Despite having successful first outcomes, we have a few limitations of the study that need to be improved in future studies. One is the high number of false-positive cases, possibly caused by other sympathetic-inducing activities reflected in EDA signals, as other sympathetic activities such as stress are known to induce changes in EDA. However, stress is a confounding factor that is present in both pre and post pain stimuli conditions. Since we are comparing the difference in TVSymp and MTVSymp values before and immediately after stimulus, the portion of the overall EDA due to stress is likely to be a non-factor. Thus, we can effectively examine the effect of pain on EDA without significant stress effect. Another limitation is that our application was not tested in the presence of motion artifacts, which can corrupt EDA signals and deteriorate the performance of the tool. For this, we will develop an approach to discriminate between stress and pain in our future studies. Moreover, our method cannot locate pain sources, which will require additional sensors (e.g., electromyography). This needs to be addressed in future studies. Finally, one of our limitations is that our application can only discriminate between the presence of pain and painless segments, as opposed to providing a pain scale. Our ultimate goal is to quantitatively detect multiple levels of pain such as low, medium, and high levels of pain.

While the purpose of the current study was to detect and quantify short-lasting acute pain, the next logical step is to investigate long-lasting chronic pain. This investigation will be more challenging as both thermal grill and electrical pulse stimuli are only designed to elicit acute pain. In addition, the chronic pain data captured by EDA device will not be as responsive and conclusive as that of acute pain. Furthermore, there will be some difficulty in separating stress from chronic pain as the latter may induce the former response. Additionally, the EDA indices analyzed using a smartphone exhibited significant differences between induced pain and no pain in the studied population. Pain perception has a difference based on one’s gender, ethnicity, and culture [50,51,52,53]. Further, different EDA signals have been observed in different genders [54]. Our study is limited to the subjects we studied so it is not clear if the results are generalizable to the overall population. Further examination of individual sensitivity of the features involving many more and diverse subjects must be studied in the future. 

## 5. Conclusions

We developed a smartphone-based system for real-time objective pain measurement using a wrist-worn electrodermal activity (EDA) device. With further improvement, noted in the Discussion section, it will help both chronic and acute pain patients that have communication disorders. The device can also be applicable to infants who are not able to communicate their pain levels.

## Figures and Tables

**Figure 1 sensors-21-03956-f001:**
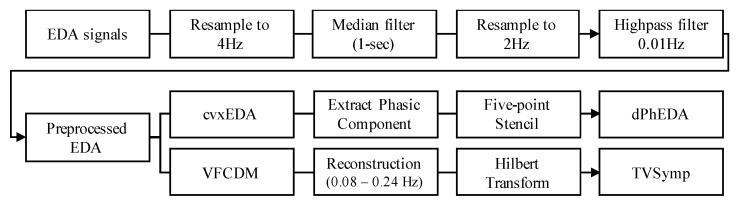
Flow chart of TVSymp and MTVSymp calculation. EDA: electrodermal activity, cvxEDA: convex EDA optimization method, VFCDM: variable frequency complex demodulation.

**Figure 2 sensors-21-03956-f002:**
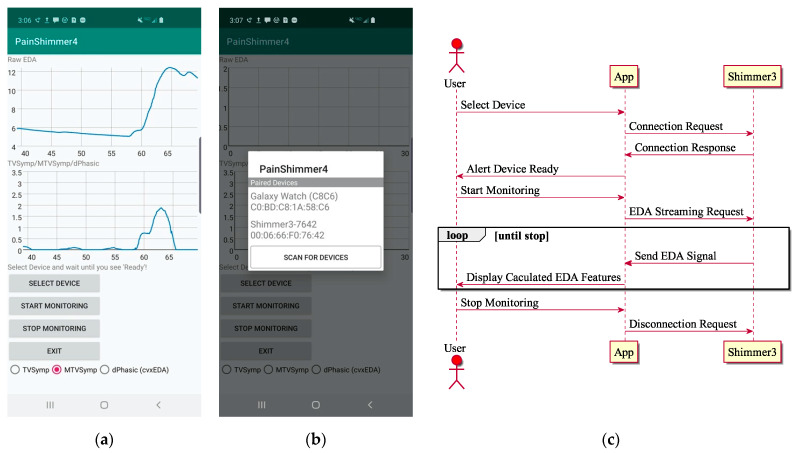
(**a**) Main screen (heat pain was induced around 56 s), (**b**) device selection in the smartphone application, and (**c**) the unified model language (UML) sequence diagram for the application.

**Figure 3 sensors-21-03956-f003:**
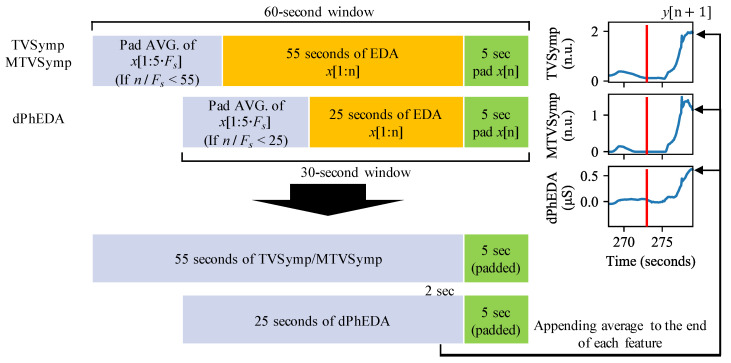
Objective real-time pain measuring scheme. The red bars in the signals represent where a stimulus was given. F_s_: sampling frequency (2 Hz).

**Figure 4 sensors-21-03956-f004:**
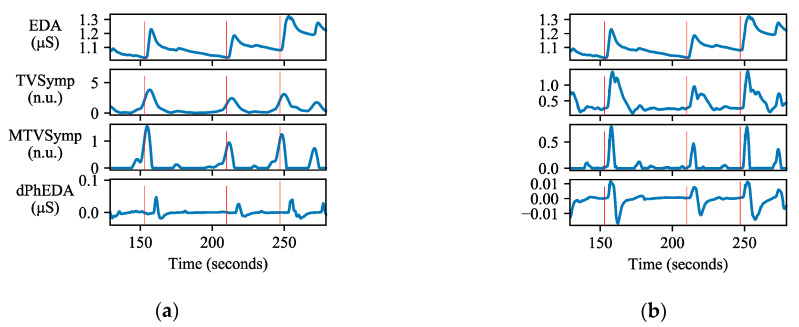
An example of EDA signal and derived time series: (**a**) EDA indices calculated using the entire recording, (**b**) EDA indices calculated using our objective real-time pain measuring scheme based on windowing. Red bars indicate where stimuli were given.

**Figure 5 sensors-21-03956-f005:**
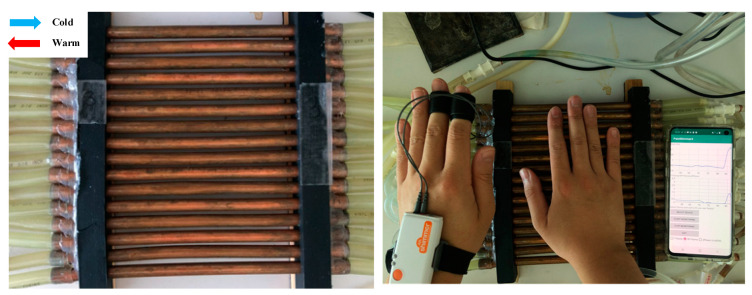
Thermal grill (**left**) and an example of data collection (**right**).

**Figure 6 sensors-21-03956-f006:**
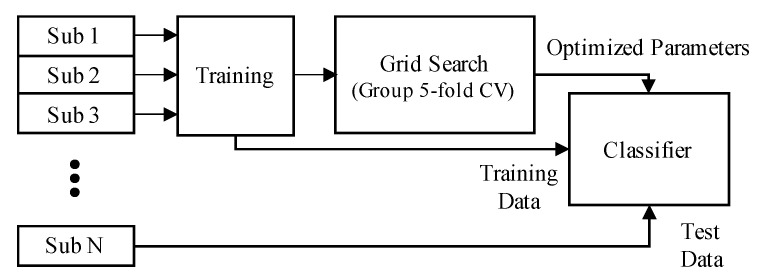
Grid-search cross-validation scheme with leave-one-subject-out cross-validation (each fold). This was repeated as many as the number of subjects to test all subjects.

**Figure 7 sensors-21-03956-f007:**
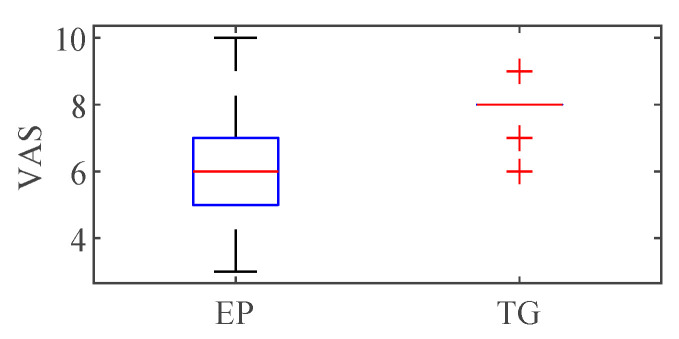
Comparison of Visual analog scale (VAS) between EP and TG stimulation.

**Figure 8 sensors-21-03956-f008:**
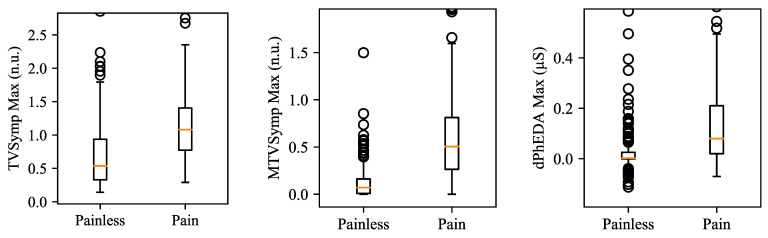
Boxplots for maximum values of TVSymp, MTVSymp, and dPhEDA for both datasets. The circles in the boxplots indicate outliers. Outliers were set if each datum is above Q1 − 1.5 × (Q3 − Q1) or below Q3 + 1.5 × (Q3 − Q1). Q1 and Q3 represent the first and third quartiles, respectively.

**Figure 9 sensors-21-03956-f009:**
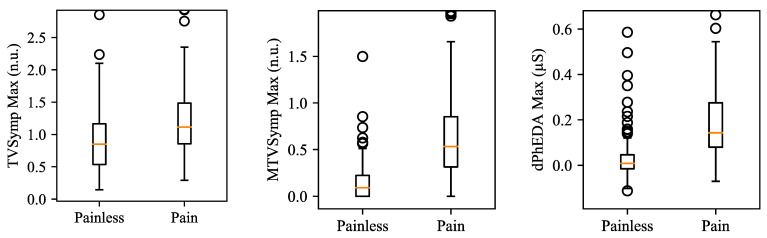
Boxplots for maximum values of TVSymp, MTVSymp, and dPhEDA for EP dataset. The circles in the boxplots indicate outliers. Outliers were set if each datum is above Q1 − 1.5 × (Q3 − Q1) or below Q3 + 1.5 × (Q3 − Q1). Q1 and Q3 represent the first and third quartiles, respectively.

**Figure 10 sensors-21-03956-f010:**
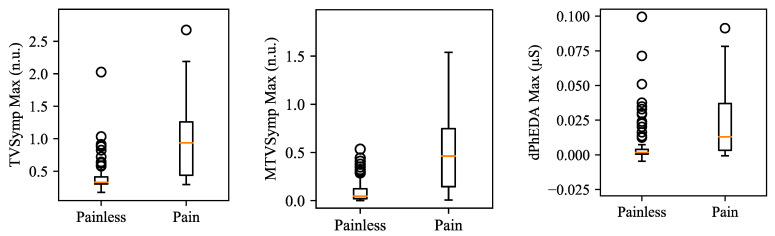
Boxplots for maximum values of TVSymp, MTVSymp, and dPhEDA for TG dataset. The circles in the boxplots indicate outliers. Outliers were set if each datum is above Q1 − 1.5 × (Q3 − Q1) or below Q3 + 1.5 × (Q3 − Q1). Q1 and Q3 represent the first and third quartiles, respectively.

**Table 1 sensors-21-03956-t001:** Parameter candidates for each classifier.

Classifiers	Parameters	Values
Support Vector Machine	C	1, 10, 100, 1000
	Gamma	0.0001, 0.0001, 0.001, 0.1
Decision Tree and Random Forest	Criterion	Gini, Entropy
Multi-layer Perceptron	Hidden Layer	1, 2, 3 (Hidden Unit: 100)
	Activation	Logistic, tanh, rectifier linear unit
	Solver	Stochastic gradient descent, Adam, LBFGS
	Learning rate	0.0001, 0.001, 0.01
Logistic Regression	Solver	Newton-CG, LBFGS, Lib Linear, SAG, SAGA
K-nearest neighbors	K	3, 5, 7, 9

Adam [42]. Newton-CG [43]. LBFGS: Limited-memory Broyden–Fletcher–Goldfarb–Shannon [44]. SAG: Stochastic Average Gradient [45]. SAGA [46].

**Table 2 sensors-21-03956-t002:** Statistical analysis on difference between painless and pain segments for both EP and TG datasets.

Features	Fisher’s Ratio		AUROC			
	Mean	Max	Mean	95% CI of Mean	Max	95% CI of Max
TVSymp	0.272	0.591	0.660	0.613–0.707	0.746	0.704–0.789
MTVSymp	0.810	0.954	0.852	0.819–0.885	0.877	0.847–0.908
dPhEDA	0.566	0.567	0.829	0.793–0.865	0.821	0.784–0.857

All features showed significant difference between painless and pain segments (*p* < 0.001, nested Ranks Test). AUROC: area under the receiver operating characteristics. CI: confidence interval.

**Table 3 sensors-21-03956-t003:** Machine-learning results for both EP and TG datasets.

Classifiers	Protocol	Accuracy (95% CI)	Sensitivity (95% CI)	Specificity (95% CI)
Support Vector Machine Linear (L-SVM)	1	0.808 (0.751–0.862)	0.752 (0.623–0.860)	0.863 (0.803–0.918)
	2	0.795 (0.739–0.851)	0.670 (0.578–0.762)	0.920 (0.867–0.973)
	3	0.818 (0.776–0.861)	0.771 (0.705–0.836)	0.866 (0.813–0.919)
Support Vector Machine 3rd order Polynomial (P-SVM)	1	0.774 (0.708–0.835)	0.813 (0.687–0.921)	0.736 (0.645–0.820)
	2	0.780 (0.723–0.837)	0.670 (0.578–0.762)	0.890 (0.829–0.951)
	3	0.729 (0.680–0.778)	0.873 (0.820–0.925)	0.586 (0.509–0.663)
Support Vector Machine Radial basis function (R-SVM)	1	0.811 (0.750–0.870)	0.755 (0.621–0.861)	0.867 (0.805–0.923)
	2	0.795 (0.739–0.851)	0.640 (0.546–0.734)	0.950 (0.907–0.993)
	3	0.815 (0.772–0.858)	0.752 (0.684–0.819)	0.879 (0.828–0.930)
Decision Tree	1	0.761 (0.706–0.812)	0.733 (0.624–0.826)	0.789 (0.728–0.849)
	2	0.625 (0.558–0.692)	0.290 (0.238–0.422)	0.960 (0.953–1.000)
	3	0.764 (0.717–0.811)	0.796 (0.733–0.859)	0.732 (0.663–0.802)
Random Forest	1	0.815 (0.754–0.869)	0.789 (0.662–0.900)	0.842 (0.784–0.896)
	2	0.655 (0.589–0.721)	0.330 (0.238–0.422)	0.980 (0.953–1.000)
	3	0.809 (0.765–0.852)	0.796 (0.733–0.859)	0.822 (0.762–0.882)
Multi-layer Perceptron (MLP)	1	0.796 (0.733–0.855)	0.759 (0.617–0.873)	0.833 (0.760–0.899)
	2	0.800 (0.745–0.855)	0.660 (0.567–0.753)	0.940 (0.893–0.987)
	3	0.701 (0.650–0.751)	0.904 (0.858–0.950)	0.497 (0.419–0.575)
Logistic Regression	1	0.813 (0.757–0.869)	0.754 (0.602–0.880)	0.873 (0.816–0.926)
	2	0.800 (0.745–0.855)	0.670 (0.578–0.762)	0.930 (0.880–0.980)
	3	0.831 (0.790–0.873)	0.866 (0.813–0.919)	0.796 (0.733–0.859)
K-nearest Neighbors (KNN)	1	0.780 (0.719–0.833)	0.799 (0.678–0.894)	0.760 (0.689–0.825)
	2	0.795 (0.739–0.851)	0.670 (0.578–0.762)	0.920 (0.880–0.980)
	3	0.806 (0.762–0.849)	0.803 (0.740–0.865)	0.809 (0.747–0.870)

Protocol 1: Both TG and EP dataset with leave-one-subject-cross-validation, Protocol 2: EP for training and TG for testing, Protocol 3: TG for training and EP for testing. CI: confidence interval.

**Table 4 sensors-21-03956-t004:** Statistical analysis on difference between painless and pain segments for EP dataset.

Features	Fisher’s Ratio		AUROC			
	Mean	Max	Mean	95% CI of Mean	Max	95% CI of Max
TVSymp	0.142	0.495	0.576	0.513–0.639	0.698	0.640–0.756
MTVSymp	0.755	0.948	0.852	0.809–0.894	0.893	0.857–0.930
dPhEDA	0.709	0.715	0.872	0.832–0.912	0.888	0.851–0.925

All features showed significant difference between painless and pain segments (*p* < 0.001, nested Ranks Test), except for TVSymp Mean (*p* = 0.0731, linear mixed effects model). AUROC: area under the receiver operating characteristics. CI: confidence interval.

**Table 5 sensors-21-03956-t005:** Machine-learning results for EP dataset.

Classifiers	Accuracy (95% CI)	Sensitivity (95% CI)	Specificity (95% CI)
Support Vector Machine Linear (L-SVM)	0.819 (0.755–0.874)	0.795 (0.668–0.890)	0.844 (0.771–0.909)
Support Vector Machine 3rd order Polynomial (P-SVM)	0.837 (0.776–0.889)	0.909 (0.857–0.955)	0.765 (0.661–0.865)
Support Vector Machine Radial basis function (R-SVM)	0.820 (0.753–0.871)	0.795 (0.663–0.888)	0.844 (0.771–0.907)
Decision Tree	0.759 (0.700–0.818)	0.758 (0.660–0.843)	0.760 (0.646–0.870)
Random Forest	0.843 (0.804–0.882)	0.877 (0.811–0.939)	0.809 (0.713–0.902)
Multi-layer Perceptron (MLP)	0.813 (0.737–0.880)	0.822 (0.679–0.926)	0.805 (0.681–0.906)
Logistic Regression	0.815 (0.739–0.873)	0.773 (0.625–0.880)	0.857 (0.777–0.924)
K-nearest Neighbors (KNN)	0.841 (0.778–0.900)	0.887 (0.794–0.959)	0.796 (0.706–0.874)

CI: confidence interval.

**Table 6 sensors-21-03956-t006:** Statistical analysis on difference between painless and pain segments for TG dataset.

Features	Fisher’s Ratio		AUROC			
	Mean	Max	Mean	95% CI of Mean	Max	95% CI of Max
TVSymp	0.693	0.991	0.849	0.795–0.903	0.859	0.806–0.911
MTVSymp	1.001	1.039	0.845	0.791–0.900	0.851	0.797–0.905
dPhEDA	0.365	0.387	0.872	0.832–0.912	0.888	0.851–0.925

All features showed significant difference between painless and pain segments (*p* < 0.001, nested Ranks Test. AUROC: area under the receiver operating characteristics. CI: confidence interval.

**Table 7 sensors-21-03956-t007:** Machine-learning results for TG dataset.

Classifiers	Accuracy (95% CI)	Sensitivity (95% CI)	Specificity (95% CI)
Support Vector Machine Linear (L-SVM)	0.740 (0.625–0.855)	0.590 (0.340–0.840)	0.890 (0.780–0.980)
Support Vector Machine 3rd order Polynomial (P-SVM)	0.765 (0.660–0.870)	0.610 (0.360–0.830)	0.920 (0.820–1.000)
Support Vector Machine Radial basis function (R-SVM)	0.735 (0.620–0.845)	0.550 (0.290–0.800)	0.920 (0.810–0.990)
Decision Tree	0.755 (0.640–0.860)	0.690 (0.480–0.870)	0.820 (0.720–0.920)
Random Forest	0.755 (0.645–0.865)	0.670 (0.400–0.900)	0.840 (0.710–0.950)
Multi-layer Perceptron (MLP)	0.750 (0.645–0.865)	0.670 (0.400–0.890)	0.830 (0.670–0.950)
Logistic Regression	0.760 (0.640–0.870)	0.640 (0.390–0.870)	0.880 (0.760–0.980)
K-nearest Neighbors (KNN)	0.735 (0.635–0.845)	0.720 (0.490–0.920)	0.750 (0.650–0.850)

CI: confidence interval.

## Data Availability

The data presented in this study are available on request from the corresponding author.

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
