# Peer review of "Real-Time High-Level Acute Pain Detection Using a Smartphone and a Wrist-Worn Electrodermal Activity Sensor"

_sensors, 2021, doi:10.3390/s21123956_

Round 1

Reviewer 1 Report

Dear Editor,

Thank you for the opportunity to re-evaluate sensors-1253052. I believe the authors did a diligent job in revising their manuscript and that they indeed covered most concerns accurately. 

Some minor textual issues remain that should be relatively easy to solve by critical rereading and may be rephrasing some sentences. I trust the authors will manage to do so. 

Author Response

Dear Reviewer 1,

We sincerely appreciate your careful reading of our manuscript and helpful comments/suggestions you have provided to improve our revised paper.  We have carefully re-read the paper and edited the text when needed.

Thank you again.

Reviewer 2 Report

Thanks to the authors for addressing the comments.

Specific comment, as in the initial review:

"Is dPhEDA really a useful parameter for this purpose? Might the authors want to reflect this in the results, discussion and conclusion?"

The authors have commented on this and addressed the issue in the paper. They argue that in previous work dPhEDA was found to provide useful information.

The question remains if the results of this work provide additional information about the value of dPhEDA in the context of acute pain.

The authors therefore might want to discuss if the results described in this paper do add any new knowledge abouth dPhEDA in the context of acute pain.

Author Response

Dear Reviewer 2,

Thank you very much for your helpful comments and suggestions to improve our paper.  Per your request, we have revised the paper and it is hoped that our revised text addressed your concern.

Thank you again.

Specific comment, as in the initial review:

"Is dPhEDA really a useful parameter for this purpose? Might the authors want to reflect this in the results, discussion and conclusion?"

The authors have commented on this and addressed the issue in the paper. They argue that in previous work dPhEDA was found to provide useful information.

The question remains if the results of this work provide additional information about the value of dPhEDA in the context of acute pain.

The authors therefore might want to discuss if the results described in this paper do add any new knowledge abouth dPhEDA in the context of acute pain.

RESPONSE: Thank you again for reviewing our paper and your helpful comments which improved our manuscript. We apologize that our previous revision was not able to address the issues properly. Our new finding of dPhEDA is that this feature was able to detect high-level thermal pain.  This was also the case for the electrical shock pain as demonstated in our previous work. We revised the related sentences in the discussion section as follows:

In our previous study, we showed that the differential characteristics of EDA signals (dPhEDA and MTVSymp) showed high sensitivity to detecting electrical pain [22]. In this paper, MTVSymp and dPhEDA exhibited good sensitivity to both thermal and electrical pain with higher Fisher’s ratio and AUROC than of TVSymp.  Hence, both MTVSymp and DPhEDA features were found to be important in detecting acute pain.